# Brief communication: Storstrømmen glacier, Northeast Greenland, primed for end-of-decade surge

Jonas K. Andersen[1], Rasmus P. Meyer[1], Flora S. Huiban[1], Mads L. Dømgaard[1], Romain Millan[2], and Anders A. Bjørk[1]

[1]Department of Geosciences and Natural Resource Management, University of Copenhagen, Copenhagen Denmark
[2]Univ. Grenoble Alpes, CNRS, IRD, INRAE, Grenoble-INP, IGE (UMR 5001), 38000 Grenoble, France

**Correspondence:** Jonas K. Andersen (joka@ign.ku.dk)

**Abstract.** Storstrømmen Glacier, a large surge-type marine-terminating glacier in northeast Greenland, is currently in a quiescent phase. We reassess the glacier's development toward a potential surge by updating time series of surface elevation, ice velocity, and grounding line location through 2023. Observations suggest the glacier is approaching pre-surge conditions, with a possible surge onset projected to occur between 2027 and 2040. Additionally, we document several lake drainage events that caused transient ice flow accelerations without triggering a surge. The findings underscore the importance of continued monitoring to improve our understanding of surge initiation mechanisms, including the influence of transient drainage events.

## 1 Introduction

Storstrømmen is a large marine-terminating outlet glacier in northeast Greenland. Together with Zachariae Isstrøm and Nioghalvfjerdsfjorden, it drains the 600 km long Northeast Greenland Ice Stream (NEGIS) and the catchment of Storstrømmen alone holds a 26 cm sea level equivalent (Mouginot et al., 2019). Storstrømmen branches out from NEGIS and flows southeast into Dronning Louise Land, where it finally terminates in Borgfjorden. At the terminus, Storstrømmen merges with L. Bistrup Bræ, which flows into Borgfjorden from the south (Figure 1a). The glaciers are both grounded on bedrock 10-15 km behind their common ice front, forming a floating ice shelf with an area of about 340 km$^2$ (as of April 2024).

Storstrømmen and L. Bistrup Bræ are among the largest surge-type glaciers in the world (Higgins, 1991; Reeh et al., 1994; Mouginot et al., 2018). A surging glacier is characterized by active phases of highly increased flow speed, typically due to increased basal sliding, separated by quiescent phases of stagnant ice flow (Kamb et al., 1985). Surging glaciers are often divided into *Alaskan* and *Svalbard* surge-type glaciers (Murray et al., 2003), based on the magnitude and duration of their active phases, though the underlying mechanisms driving the surge cycle may be similar for both groups (Benn et al., 2019). For the Alaskan type, glaciers exhibit a sudden and sharp flow acceleration, often reaching flow speeds of many km/y, before suddenly stagnating back to pre-surge speeds. Conversely, Svalbard surge-type glaciers generally exhibit a more gradual surge onset and termination, with flow speed acceleration/deceleration often occurring over several years (Dowdeswell et al., 1991). Storstrømmen and L. Bistrup both belong to the Svalbard surge-type, as past observations indicate an active phase onset/termination spanning several years (Reeh et al., 1994; Mouginot et al., 2018). Based on past field expeditions and remote sensing

data, Mouginot et al. (2018) inferred active surge phases of Storstrømmen centered around 1910 and 1982, suggesting a surge cycle periodicity of about 70 years. During the latest Storstrømmen surge, ice velocities reached 2-3 km/y, the ice front advanced more than 10 km, and the estimated ice discharge at the front of Storstrømmen (during 1975-1988) was $126 \pm 14$ Gt (Mouginot et al., 2018), about half the annual discharge of Zachariae Isstrøm and Nioghalvfjerdsfjorden combined (Mankoff et al., 2020). For L. Bistrup, surges were inferred during approximately 1912, the 1950s, and 1993, suggesting a surge periodicity on the order of 30-50 years. During the latest surge of L. Bistrup, discharge rates reached only $\sim 10\%$ of those observed for Storstrømmen (Mouginot et al., 2018).

Storstrømmen and L. Bistrup are currently in a quiescent phase, with both glaciers exhibiting low average flow speeds ($< 200$ m/y) and limited seasonal fluctuations. In contrast to non-surge-type glaciers in Greenland, for surging glaciers in the quiescent phase flow speeds decrease up-glacier. For Storstrømmen and L. Bistrup, ice is nearly stationary (average flow $< 20$ m/y) in the last 30-50 km before the ice front. During the current quiescent phase, Mouginot et al. (2018) showed that the upper part of Storstrømmen (30-70 km from the grounding line) has been thickening, while the downstream region (0-30 km from the grounding line) has been thinning, causing an upstream build-up of ice. Meanwhile, the grounding line of Storstrømmen was shown to retreat at a steady rate. Through these observations, Mouginot et al. (2018) found that Storstrømmen was steadily approaching the surface elevation and grounding line conditions of 1978, the approximate time of the latest surge onset. By a simple linear projection, it was concluded that these pre-surge conditions would be met in 2027-2030. Additionally, more recent studies have noted a further grounding line retreat of Storstrømmen (Millan et al., 2023; Rignot et al., 2022).

While the surge cycle of Storstrømmen Glacier has been extensively studied, the precise trigger mechanism remains uncertain. In this study, we extend the work of Mouginot et al. (2018) by updating time series of elevation change, ice velocity, and grounding line location through 2017-2024. Additionally, we document new observations of episodic meltwater drainage events that transiently but significantly impact ice flow, offering insights into their potential role as a trigger mechanism for future surges.

## 2 Materials and Methods

### 2.1 PROMICE ice velocity mosaics

To monitor ice velocity in the region during recent years, we use 24-day averaged 2D horizontal ice velocity mosaics from PROMICE (Solgaard et al., 2021; Solgaard and Kusk, 2024). The mosaics are generated using intensity-tracking of Synthetic Aperture Radar (SAR) measurements from the EU Copernicus Sentinel-1 satellites, utilizing all available 6- and 12-day image pairs. The mosaics have a spatiotemporal sample spacing of 200 m and 12 days and estimated velocity errors on the order of 25 m/y (Solgaard et al., 2021). We use all available mosaics for the period January 2016 to December 2023. To reduce noise in the velocity measurements, we perform a pixel-wise averaging of mosaics in 3-month periods (December-February, March-May, June-August, and September-November).

## 2.2 Double-difference SAR interferometry

We use Differential Synthetic Aperture Radar Interferometry (DInSAR) and measurements from Sentinel-1 to infer glacier grounding line location and transient changes in ice dynamics. DInSAR measures phase change between two subsequent image acquisitions. Once the effects of the satellite viewing geometry and local topography is accounted for (using an external Digital Elevation Model), the unwrapped phase is proportional to displacements in the radar line-of-sight direction (Massonnet et al., 1993). As the line-of-sight vector is slanted towards ground it has both a horizontal and a vertical component. When differencing two DInSAR retrievals, a method known as double-difference interferometry, the obtained phase changes are proportional to displacement changes between the two retrieval periods. If horizontal surface velocity is unchanged, only phase changes related to vertical displacements remain. Over floating ice shelves, double-difference measurements generally show non-zero phase changes induced by a difference in tidal amplitude between the two retrieval periods. Delineating the inland limit of the inferred vertical displacement (see Figure S1), provides an estimate of the glacier grounding line, i.e. the border that separates grounded ice from the floating ice shelf. Double-difference interferometry is a well-established method and widely regarded as the most accurate way to remotely track glacier grounding line location (Rignot, 1996; Joughin et al., 2010).

We generate interferograms using all Sentinel-1 images during January 2015 to February 2024 from one ascending and one descending track (tracks 74 and 170), following the processing approach outlined in (Andersen et al., 2020). When both satellites are available (September 2016 to December 2021) we form interferograms with a 6-day temporal baseline – otherwise, the temporal baseline is 12 days. A time series of double-difference interferograms is then generated by differencing all temporally neighbouring interferograms. We use this time series both to manually delineate the grounding line location for Storstrømmen and L. Bistrup and to monitor transient changes in ice dynamics in different parts of the study region. To estimate grounding line retreat, we draw a transect that follows a central flowline on each glacier and record the intersection point with each delineated grounding line. Additionally, we use the 1992 and 1996 grounding lines, based on ERS-1/2 data, generated by Mouginot et al. (2018).

## 2.3 Elevation change measurements

To track the evolution of ice surface elevation, we use data from NASA's Airborne Topographic Mapper (ATM) (Krabill, 2014), the multi-temporal ArcticDEM produced by the Polar Geospatial Center (Porter et al., 2022), and the 25-m AeroDEM from 1978 derived from aerial imagery (Korsgaard et al., 2016). We extract the ATM along-track lidar elevation measurements from NASA's Operation IceBridge flight campaigns of 1994, 1999, 2007 and 2014 as well as photogrammetry-based ArcticDEM strips, for every year between 2012-2023. Both the AeroDEM and the ArcticDEM strips are co-registered to the ArcticDEM mosaic, using bedrock as reference (Nuth and Kääb, 2011). We estimate an average uncertainty of the ArcticDEM elevations of 0.78 m and assign an uncertainty of 0.10 m for ATM measurements (Brunt et al., 2021).

Like Mouginot et al. (2018), we want to monitor the approach to pre-surge conditions due to ice thickening upstream and ice thinning downstream by calculating the net elevation change rate in both the upper and lower parts of Storstrømmen. We do

this by computing the mean elevation change in a section of five kilometres along the ATM flight track, in both the upper/lower Storstrømmen regions, relative to the surface elevation in 1978 (see orange sections of the flight track in Figure 1a). The 5 km
sections are centered on the points used to extract elevation in Mouginot et al. (2018). Finally, net thickening/thinning rates are computed through weighted least squares regression.

## 3 Results

Figure 1c shows relative elevation change in the Storstrømmen upper and lower zones (i.e., the sections outlined in Figure 1a) as well as grounding line location, all with respect to 1978 (pre-surge) conditions. Compared to Mouginot et al. (2018),
the elevation trend in the lower (i.e. thinning) zone has changed from $-1.4$ m/y to $-1.62 \pm 0.03$ m/y while the trend in the upper (thickening) zone remains similar (1.0 m/y vs. $1.05 \pm 0.07$ m/y), when incorporating 2017-2023 data. Extrapolating from these trends points to both zones reaching pre-surge conditions in the year 2027. However, we note that the thickening rate has decreased in the recent decade to an estimated $0.60 \pm 0.15$ m/y (during 2012-2023). At that rate, the upper zone will reach pre-surge conditions in 2040 (grey dashed line in Figure 1c). Since 2017, the Storstrømmen grounding line has continued its
retreat at a remarkably constant rate ($392 \pm 5$ m/y for 1992-2024, based on all 217 double-difference interferograms, compared to 393 m/y estimated by Mouginot et al. (2018) for the period 1992-2017, based on just 4 retrievals). Inferred grounding line locations for both Storstrømmen and L. Bistrup are shown in Figure 1b. Extrapolating the current trend in grounding line retreat, we find that the pre-surge location will be reached during 2027.

Figure 2 shows ice velocity anomalies for 3-month periods during 2016-2023 in the Storstrømmen and L. Bistrup area. For
both glaciers, ice flow is observed to be quite stable. An apparent re-occurring speed-up is observed during summer, however, as measurement noise drastically increases during this period (likely due to enhanced surface melt), the confidence in this signal is reduced. Additionally, we observe accelerated ice flow in a large region of upstream Storstrømmen outside the melt season (September 2018 to May 2019, and September 2022 to May 2023). Through the generated Sentinel-1 double-difference interferograms (described in section 2.2), we identified multiple apparent lake drainage events as likely sources of these flow
accelerations.

Figure 3a-c documents the dynamical response following the apparent drainage of an ice-dammed lake during October 2018, as well as the drainage of one or more supraglacial lakes in the upstream parts of Storstrømmen during November 2018. The double-difference interferogram phase is sensitive to changes in displacement in the radar line-of-sight, and the measurements likely show a combination of horizontal flow acceleration and vertical displacement of the ice, i.e., uplift as
water enters the subglacial system followed by subsidence once it is evacuated (see Figs. S2-S3). The downstream propagation of the dynamic response, taken to indicate the propagation of water through the subglacial system, is similar to observations from past studies (Andersen et al., 2023; Maier et al., 2023). Interestingly, this propagation does not reach the Storstrømmen grounding line but ends some 25 km upstream of it (Figure 3c). A similar pattern is observed following the drainage of one or more supraglacial lakes in December 2022 (Figure 3d-f), although in this case, we do observe signs of uplift propagating
all the way to the grounding line (Figure 3e). Inspecting Sentinel-1 amplitude images reveals local surface changes occurring

over lakes just upstream of the observed transient flow responses (Figure S4), and elevation change maps from differencing of individual ArcticDEM strips show substantial drops in elevation over the same lakes (Figure S5), prompting us to infer their drainage. Figure S6 documents additional apparent drainage events occurring in the same parts of upstream Storstrømmen during December 2018 and May 2019, co-incident with the prolonged ice flow speed-up (Figure 2, blue rectangle). It is possible that additional, undetected drainage events occur during September 2022 to May 2023 (Figure 2, red rectangle), as only 12-day Sentinel-1 image pairs are available, drastically reducing coherence and temporal resolution of the interferometric measurements. Figures S7-S8 document a drainage event in upstream L. Bistrup during January 2019. During the drainage events documented in Figure 3, we estimate ice flow speed-ups on the order of 50-120 m/y, corresponding locally to anomalies of 50-500% of average annual flow speeds (Figure 3g-h). In all cases, however, the acceleration is transient, stretching over just a few weeks to months before ice flow returns to nominal values (Figures 2 and 3). Also, the observed flow speed-ups do not appear to reach the Storstrømmen grounding line.

## 4  Discussion and conclusions

The latest surge of Storstrømmen led to a notable ice discharge ($126 \pm 14$ Gt over the active phase), which was found to approximately equal the total mass accumulated over the glacier basin in the preceding 70-year quiescent phase (Mouginot et al., 2018), suggesting a zero net glacier mass balance over a full surge cycle.

Lack of both elevation and grounding line measurements from the late 1980s means that the pre-surge conditions cannot be properly established for L. Bistrup Bræ (Mouginot et al., 2018), however, we observe an average grounding line retreat of $109 \pm 26$ m/y over the period 2015-2024 (Figure S9). The L. Bistrup grounding line is currently upstream of its location in 1992, when the glacier was a few years into the active phase of its latest surge.

Rignot et al. (2022) found that the floating sector of Storstrømmen and L. Bistrup is protected from warm ocean water intrusions and that the recent grounding line retreat is explained solely by glacier thinning, which is in line with our observation of a steady, near-constant grounding line retreat. Additionally, we observed no obvious indications of seawater intrusions in our DInSAR measurements, such as those recently observed at Thwaites glacier (Rignot et al., 2024) and inferred under Petermann glacier (Ehrenfeucht et al., 2024).

The recent (2012-2023) elevation change measurements indicate a decreased ice thickening rate in the upper Storstrømmen region compared to the prior two decades ($0.60 \pm 0.15$ m/y vs. 1.0 m/y) and an increased thinning rate in the lower regions ($-1.67 \pm 0.12$ m/y vs. $-1.4$ m/y). This is in line with recent observations of an increase in average runoff for Storstrømmen and L. Bistrup (Millan et al., 2023). Compared to Mouginot et al. (2018), we thus predict that the Storstrømmen grounding line location and lower zone elevation will meet pre-surge (1978) conditions around year 2027 (agreeing well with previous estimates), while mass build-up in the upper reservoir will likely occur later (around year 2040 vs. the previous estimate of 2029-2030), assuming a continuation of current trends (Figure 1c). A presumed requirement for surge initiation is an ice mass imbalance between the upper and lower reservoirs of Storstrømmen (Reeh et al., 1994; Mouginot et al., 2018). Although thickening in the upper reservoir has recently decreased, thinning in the lower zone and retreat of the grounding line appear

to persist at steady rates, resulting in a continuous increase in driving stress. Thus, while the precise pre-surge conditions of 1978 are unlikely to be fully reestablished by 2027, surge initiation is anticipated to be more probable in the earlier part of the 2027–2040 time frame. Inferring the timing of a coming surge would provide a valuable opportunity for acquiring in-situ and remote observations in the years up to, during, and after a glacier surge.

Using interferometric satellite radar measurements from the past decade, we find evidence of multiple supraglacial and ice-dammed lake drainages, showing that high inputs of water are regularly provided to the subglacial environment. The drainage events all occur outside the melt season, when we would generally expect a less efficient subglacial drainage system and thus a greater increase in basal water pressure, but lead only to transient flow accelerations over timescales of weeks to months. Within the general theory of glacier surges, meltwater inputs to, and subsequent changes in, the subglacial drainage system have frequently been linked to surge initiation (Kamb et al., 1985; Lingle and Fatland, 2003; Dunse et al., 2015; Haga et al., 2020). In a recently proposed generalized surge model based on enthalpy balance, an influx of water to the subglacial system is associated with an increase in enthalpy (Benn et al., 2019, 2022). While the rapid drainage events presented here clearly did not initiate a surge for either glacier, it is possible that similar events may contribute to future surge initiation, once the pre-surge configuration, and thus a state of mass/enthalpy imbalance, has been reached. Alternatively, the external forcing from these episodic, transient inputs of meltwater to the glacier bed may play a lesser role in initiating surges of Storstrømmen and L. Bistrup, which instead may be controlled by a more gradual evolution in basal water pressure and subglacial drainage configuration.

A common theory is that surge initiation occurs once enough basal water is accumulated to raise water pressure above ice overburden pressure, enhancing basal motion through sliding and commencing a velocity-frictional heating feedback (Clarke, 1976; Benn et al., 2019). Our observations indicate downstream propagation of water through the subglacial system over timescales of weeks to months, however, it is unclear how much (if any) of this water is stored in the subglacial system. We do note that for several of the identified drainage events, downstream propagation of subglacial water appeared to cease 25 km upstream of the Storstrømmen grounding line (Figs. 3a-c, Fig. S6), suggesting that the drained water volume might not have been fully evacuated. Investigating similar surface-to-bed drainage events (including their frequency) in the time up to and during the next Storstrømmen surge may reveal detailed changes in the subglacial drainage system (in the form of spatial uplift/subsidence patterns - see Figs. S2-S3 - and the temporal propagation of the dynamic response). Continued close monitoring of hydrology-dynamical effects could then help establish the impact of supra- and subglacial drainage events on the surge cycles of Storstrømmen, L. Bistrup Bræ, and other surge-type glaciers.

*Code and data availability.* Ice velocity measurements from PROMICE are available at https://dataverse.geus.dk/dataverse/Ice_velocity. NASA ATM data was downloaded from https://nsidc.org/data/ILATM2. ArcticDEM strips are available through https://www.pgc.umn.edu/data/arcticdem/. Sentinel-1/2 imagery is available at https://scihub.copernicus.eu/. The generated collection of grounding lines will be made available in a public repository, along with code for generating Figures 1-3, upon acceptance of the manuscript

*Author contributions.* J.K.A. and A.A.B designed the study. J.K.A, R.P.M, F.S.H, and M.L.D performed data processing, with analysis contributions from all authors. J.K.A wrote the initial draft of the manuscript, with editing from all other authors.

*Competing interests.* The authors declare no competing interests.

*Acknowledgements.* The authors acknowledge support from the Villum Foundation (Villum Young Investigator grant no. 29456) and the
190 Independent Research Fund Denmark Sapere Aude Research Leader (Grant 10.46540/2064-00050B). We thank reviewers Laurence Gray and Adrian Luckman as well as editor Stephen Livingstone for insightful comments that greatly improved the paper. Additionally, we thank John Merryman Boncori, Anders Kusk, and Anne Solgaard for helpful discussions.

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

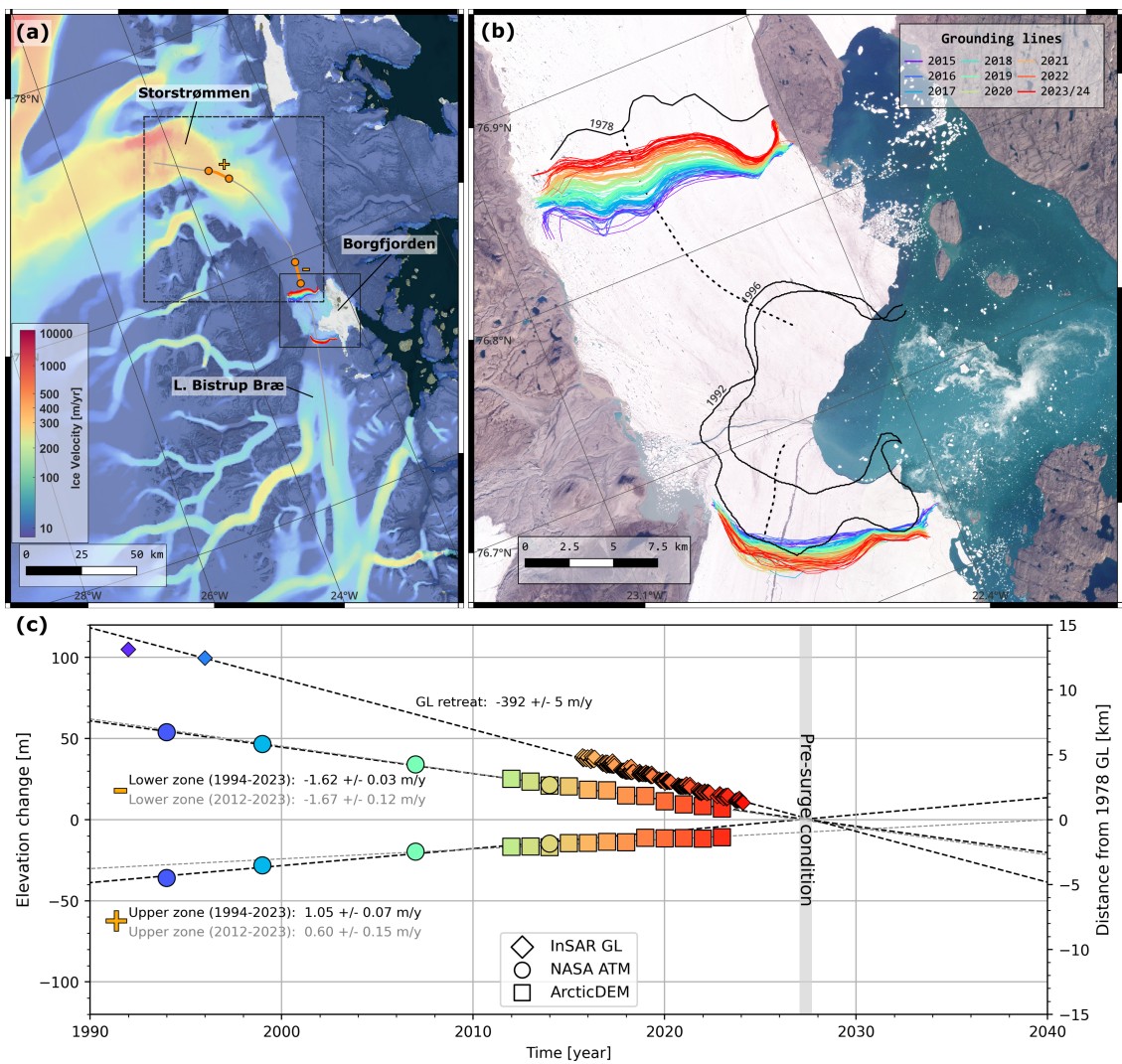

**Figure 1.** (a) Overview of Storstrømmen and L. Bistrup glaciers. Background shows PROMICE 2016-2023 average ice velocity (Solgaard et al., 2021). Solid grey line shows ATM flight track, with orange sections denoting regions of estimated elevation change. Solid black rectangle indicates boundaries of panel (b) and the dashed black rectangle shows boundaries of Figure 3. (b) Glacier fronts of Storstrømmen and L. Bistrup, with grounding lines indicated by solid lines (black lines are from Mouginot et al. (2018), while colored lines are from this study). The dashed black lines show transects used to evaluate grounding line location change. Background image is a PlanetScope composite from 18th August 2023 (Planet Labs PBC, 2024). (c) Time series of elevation change (circles and squares) in upper and lower zones of Storstrømmen (orange sections labelled + and −, respectively, in panel (a)) with respect to the 1978 DEM, extracted from both NASA ATM and ArcticDEM measurements. Diamonds indicate distance of the Storstrømmen grounding line from its 1978 position, evaluated along the dashed transect in (b). Grounding line locations are measured with Sentinel-1 (2015-2024) and ERS-1/2 (1992-1996, obtained from Mouginot et al. (2018)). Dashed black/gray lines indicate estimated trends (with 95% confidence intervals provided in the adjacent text). The gray box indicates the timing at which grounding line and surface elevation conditions match those of 1978, when the last Storstrømmen surge was initiated.

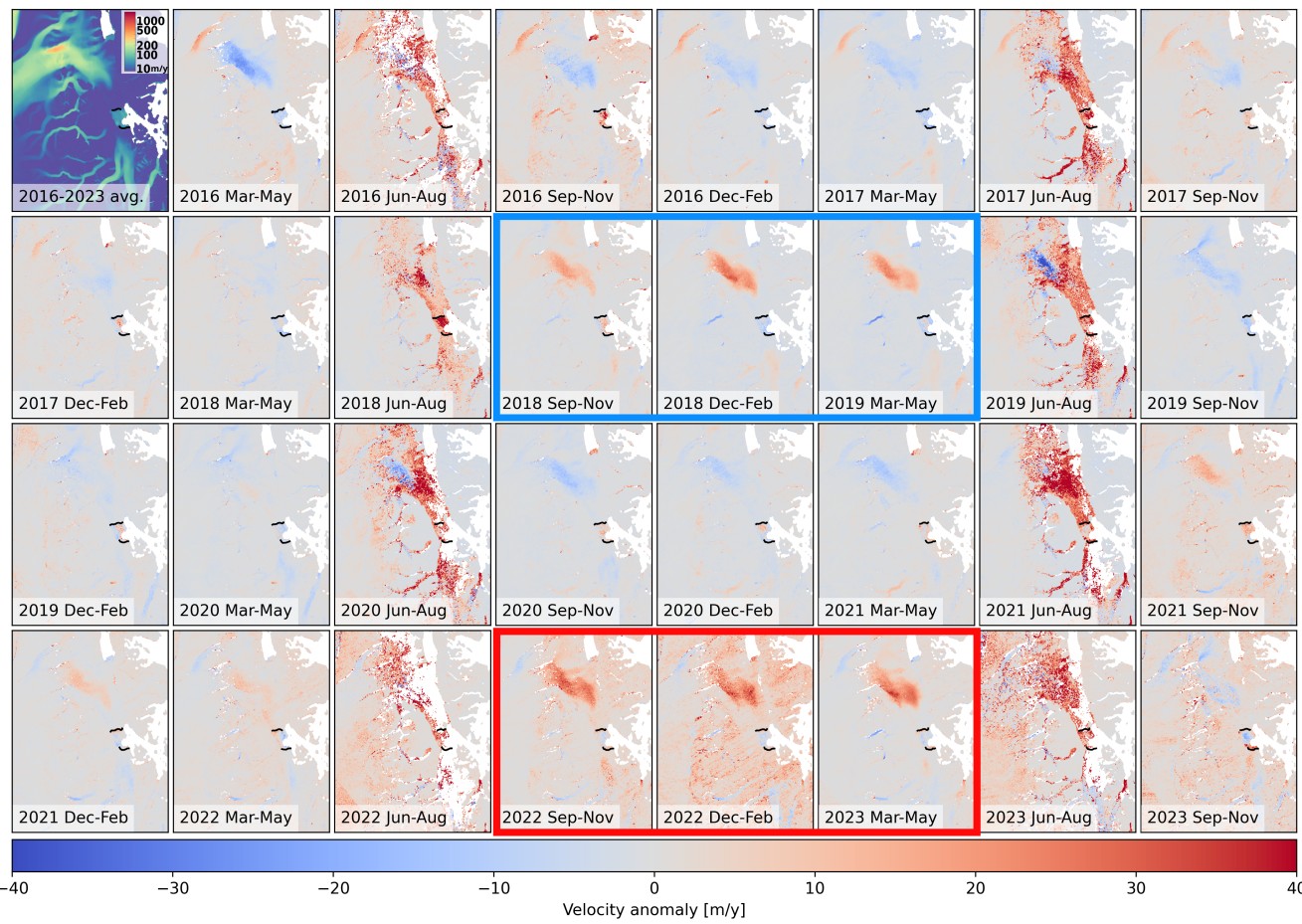

**Figure 2.** Time series of PROMICE ice velocity magnitude anomalies for 3-month periods during 2016-2023, with respect to the pixel-wise median of the full time series. Top left panel shows 2016-2023 median velocity magnitude (in m/y). Solid black lines indicate grounding lines. Panels surrounded by the blue and red borders highlight flow anomalies related to inferred drainage events (further documented in Figure 3).

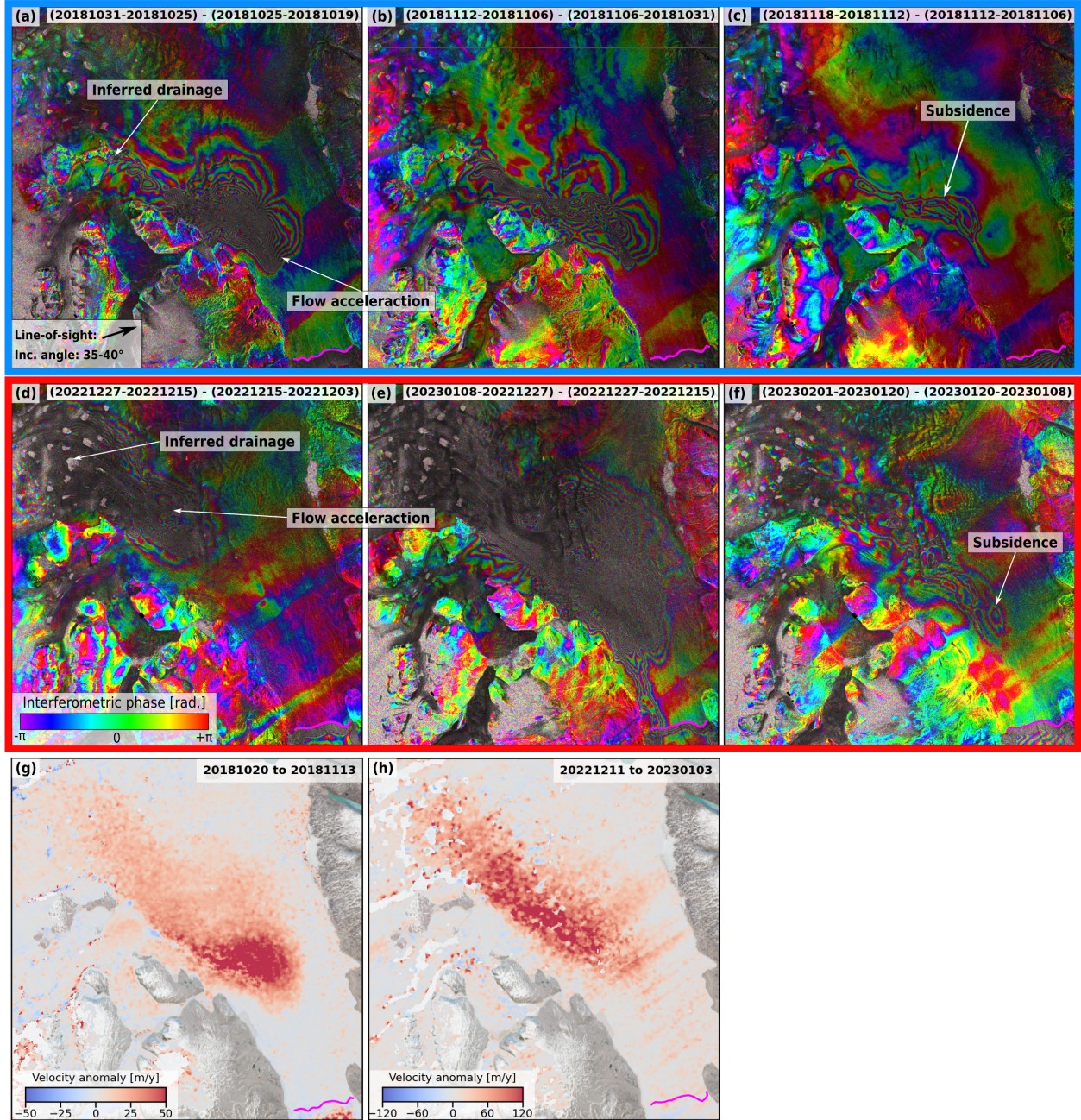

**Figure 3.** Sentinel-1 double-difference interferograms showing dynamical effects related to apparent drainage events in upstream Storstrømmen during fall 2018 (panels (a)-(c)) and winter 2022/2023 (panels (d)-(f)). The imaged region is indicated by the dashed black rectangle in Figure 1 and panel (a) indicates the ground-projected line-of-sight direction and incidence angle of Sentinel-1 track 74 (used for all the measurements in (a)-(f)). The dense fringe patterns indicate changes in relative motion (in the direction of the radar antenna). Panels (g) and (h) show PROMICE ice velocity magnitude anomalies for two 24-day periods spanning the identified drainage events (masked using the GIMP classification mask (Howat et al., 2017) and overlaid on a Sentinel-2 optical image). The solid magenta line indicates the Storstrømmen grounding line.