# Peer review of "Brief communication: Storstrømmen glacier, Northeast Greenland, primed for end-of-decade surge"

_EGUsphere, 2024_

## Referee Comment (RC1)

In their TCD brief communication: 'Storstrømmen glacier, Northeast Greenland, primed for end-of-decade surge' Andersen et al., use ice velocity, double difference interferometric SAR, and DEM data to study the state of two large surge glaciers in the north-east of the Greenland Ice Sheet. The work builds on the earlier work of Mouginot et al., (2018) who reviewed the surge history of the two glaciers and used recent and historical data to suggest that the apparent conditions for the next surge of the Storstrømmen Glacier could be met in the 2027 – 2030 time period. Andersen et al., added recent data, from 2017 to 2023, and exploited all the height change data to suggest that the next surge of this glacier could occur in the 2027 – 2040 period. They also use some double difference interferograms to show change in the line-of-sight ice surface position over particular time periods and suggest that these changes present evidence for lake drainage events leading to changes in ice movement.

My review is in three parts. Firstly, I try to answer the questions suggested by TC in their guidance to reviewers, secondly, I add some detailed comments on the text and figures, and thirdly present a summary of my thoughts on the paper.

   1.  *Does the paper address relevant scientific questions within the scope of TC?*

Yes.

   2.  *Does the paper present novel concepts, ideas, tools, or data?*

The recent ice velocity, double difference interferometry and elevation change data included in the paper are new and of interest in the context of when the two glaciers may surge next. However, the methodology is not unique.

   3.  *Are substantial conclusions reached?*

The answer to this question is inevitably somewhat subjective. Remembering that the submission is in the form of a 'brief communication' my response to this question is that the results and conclusions in the communication are sufficiently useful that they can be considered as 'substantial'.

   4.  *Are the scientific methods and assumptions valid and clearly outlined?*

Yes.

   5.  *Are the results sufficient to support the interpretations and conclusions?*

Yes, although I think that some of the discussion and figure labels on 'inferred drainage', 'subsidence' and 'flow acceleration' should be qualified.

6. *Is the description of experiments and calculations sufficiently complete and precise to allow their reproduction by fellow scientists (traceability of results)?*

Yes. But as outlined below some of interpretation could benefit from improved figures and methods.

7. *Do the authors give proper credit to related work and clearly indicate their own new/original contribution?*

Yes.

8. *Does the title clearly reflect the contents of the paper?*

Yes.

9. *Does the abstract provide a concise and complete summary?*

Yes.

10. *Is the overall presentation well structured and clear?*

Yes.

11. *Is the language fluent and precise?*

Yes.

12. *Are mathematical formulae, symbols, abbreviations, and units correctly defined and used?*

There are suggested changes to the use of the terms 'accumulation' and 'ablation' rates.

13. *Should any parts of the paper (text, formulae, figures, tables) be clarified, reduced, combined, or eliminated?*

There are suggested improvements to some of the figures, see below.

14. *Are the number and quality of references appropriate?*

Yes.

15. *Is the amount and quality of supplementary material appropriate?*

Yes, but some improvements are suggested.

More detailed comments.

Lines 30-31.
*"Contrary to all other marine-terminating glaciers... decreasing distance to the ice front".*
I'm not sure that this is always correct, so unless you have a credible reference for the statement, I would suggest that you simplify the sentence to just describe the situation for the two glaciers.

Re 2.2 'Double-difference interferometry.' Note that any interferometric technique is by definition 'differential'.

Lines 55 – 56. *"DInSAR measures phase change between two image acquisitions, which is proportional to displacements in the radar line-of-sight direction."* This is strictly not correct; I would prefer something like the following...

Interferometric SAR (InSAR) allows the estimation of the phase change between two image acquisitions which, when the phase is unwrapped (ref), provides data on the difference in slant range to the registered pixels. When the unwrapped phase is then corrected for the different viewing positions and terrain topography (using the baseline and a DEM, add another reference), then the change in the unwrapped phase is proportional to the change in the surface displacement in the radar line-of-sight direction.

I don't think using 'DDInSAR' is necessary. 'DInSAR' is now often used for double differencing interferograms. e.g., Wild et al., 'Differential interferometric synthetic aperture radar for tide modelling in Antarctic ice-shelf grounding zones', The Cryosphere, 13, 3171–3191, https://doi.org/10.5194/tc-13-3171-2019, 2019.

Line 65. I suggest you drop 'DInSAR'... 'We generate interferograms using ...'

And line 68; e.g., 'A time series of double difference interferograms is then generated by differencing all temporally neighbouring interferograms.'

2.3 Elevation change measurements.

Line 83. Here you refer to 'accumulation rate' and 'ablation rate'. These are well defined parameters in glaciology neither of which account for height change due to ice motion (in particular along-flow strain rates). So, I suggest you change the wording here, and elsewhere, to make sure that readers understand you are referring to surface height change and height change rate in two areas, one thickening and the other thinning. In fact, I doubt that the elevation of the upper area is high enough to be in the true 'accumulation zone', i.e., with a positive surface mass balance.

3. Results

You have used two 5 km sections of the NASA ATM flight track over Storstrømmen to monitor height change in what you describe as in the 'accumulation' and ablation' zones. Here you appear to follow the work of Mouginot, et al., who also used this data. Is the 5 km section the same as used in the Mouginot work? Maybe add a comment or reference?

Line 100. The noise in the velocity anomaly sub-images for the summer periods is so large that I think you may want to rethink the estimate of summer speed up in... '... *ice flow is observed to be quite stable, except for a reoccurring summer speed-up on the order of < 40 m/y (although it should be noted that measurement noise generally increases during summer, likely due to increased surface melt).*'

Lines 105 – 112. The text here is a mixture of results and conjecture as to the cause of the observed changes. I would prefer that the conjecture re lake drainage, surface speed up based on phase, subsidence, etc., go in the next section. Some more results or clear evidence showing that there are 'lakes' which could periodically release water would be helpful (see below for one example).

Lines 120 – 123. '*... ice flow accelerations on the order of 50 – 120 m/y,*' The units are not appropriate for acceleration, do you mean change in speed of ....

Lines 151 – 154. While I think these statements are correct, the evidence is at best indirect. I think this should be acknowledged.

Figures.

Figure 1. Good.

Figure 2. The top left sub-image uses a linear colorbar. A non-linear colormap, like Figure 1a would be preferable. A vector showing the zero-Doppler or line-of-sight direction would also be useful here.

Figure 3. Panels 3c and 3f include labels indicating areas with 'subsidence'. The arguments in the text are somewhat speculative and not proven. I suggest change to 'possible subsidence'.

Supporting supplementary material.

There are 6 images in the supplementary material. The first three are in support of the text and the interpretation of the results. Figure S4 and S5 were intriguing and led to some investigation using date-specific ArcticDEMs. These supported the interpretation in the communication but provided much more convincing evidence that the glacier speed-up in Fig. S4e was due to significant water outflow from what appears to be a subglacial lake.

[Figure]

This image is from a shaded relief file for the PGC ArcticDEM from 23 March 2019. The red arrow points to the position where the ice surface dropped due to subglacial lake drainage.

Note the position of the arrowhead in Fig. S4a pointing to the area of 'inferred drainage' is not accurate and should be changed to point to the correct area.

[Figure]

This figure illustrates the DEM from 23 March 2019 after the water outflow from the subglacial lake.

[Figure]

This illustrates the height difference between pre- and post-outflow DEMs. The surface falls by up to around 85 m. This complements your observation that the SAR image brightness increased significantly between 11 and 17 Feb. 2019 as the ice broke up due to the surface collapse as the subglacial lake drained.

Presumably, the water outflow facilitates the surface ice speed-up seen in S4e and S4f.

Summary.

The brief communication contains interesting material which warrants publication. However, I would like the authors to consider the comments above, and the suggestion to use some time-specific ArcticDEMs. You are welcome to use the above example in your paper, I can provide the files, or you can download e.g., ...

SETSM_s2s041_WV01_20190323_1020010082E60600_1020010083934E00_2m_lsf_seg1.tar.gz

From the PGC directory ...

https://data.pgc.umn.edu/elev/dem/setsm/ArcticDEM/strips/s2s041/2m/n76w025/

I think the other two files I used are also in this directory (N76W25).

---

## Author Comment (AC1)

Dear Laurence Gray,

Thank you very much for your detailed review, which provided constructive comments that improved the quality of the manuscript, as well as interesting findings from ArcticDEM data. Please find below your original comments in black text and our responses in blue (quotes from the updated manuscript are in *blue italics*).

In their TCD brief communication: 'Storstrømmen glacier, Northeast Greenland, primed for end-of-decade surge' Andersen et al., use ice velocity, double difference interferometric SAR, and DEM data to study the state of two large surge glaciers in the north-east of the Greenland Ice Sheet. The work builds on the earlier work of Mouginot et al., (2018) who reviewed the surge history of the two glaciers and used recent and historical data to suggest that the apparent conditions for the next surge of the Storstrømmen Glacier could be met in the 2027 – 2030 time period. Andersen et al., added recent data, from 2017 to 2023, and exploited all the height change data to suggest that the next surge of this glacier could occur in the 2027 – 2040 period. They also use some double difference interferograms to show change in the line-of-sight ice surface position over particular time periods and suggest that these changes present evidence for lake drainage events leading to changes in ice movement. My review is in three parts. Firstly, I try to answer the questions suggested by TC in their guidance to reviewers, secondly, I add some detailed comments on the text and figures, and thirdly present a summary of my thoughts on the paper.

1. Does the paper address relevant scientific questions within the scope of TC?
Yes.

2. Does the paper present novel concepts, ideas, tools, or data?
The recent ice velocity, double difference interferometry and elevation change data included in the paper are new and of interest in the context of when the two glaciers may surge next. However, the methodology is not unique.

3. Are substantial conclusions reached?
The answer to this question is inevitably somewhat subjective. Remembering that the submission is in the form of a 'brief communication' my response to this question is that the results and conclusions in the communication are sufficiently useful that they can be considered as 'substantial'.

4. Are the scientific methods and assumptions valid and clearly outlined?
Yes.

5. Are the results sufficient to support the interpretations and conclusions?
Yes, although I think that some of the discussion and figure labels on 'inferred drainage', 'subsidence' and 'flow acceleration' should be qualified.
Please see our responses to your comments under "Figures".

6. Is the description of experiments and calculations sufficiently complete and precise to allow their reproduction by fellow scientists (traceability of results)?

Yes. But as outlined below some of interpretation could benefit from improved figures and methods.

7. Do the authors give proper credit to related work and clearly indicate their own new/original contribution?
Yes.

8. Does the title clearly reflect the contents of the paper?
Yes.

9. Does the abstract provide a concise and complete summary?
Yes.

10. Is the overall presentation well structured and clear?
Yes.

11. Is the language fluent and precise?
Yes.

12. Are mathematical formulae, symbols, abbreviations, and units correctly defined and used?
There are suggested changes to the use of the terms 'accumulation' and 'ablation' rates.
These terms have been changed - please see our response under your comment titled "2.3 Elevation change measurements" below.

13. Should any parts of the paper (text, formulae, figures, tables) be clarified, reduced, combined, or eliminated?
There are suggested improvements to some of the figures, see below.

14. Are the number and quality of references appropriate?
Yes.

15. Is the amount and quality of supplementary material appropriate?
Yes, but some improvements are suggested.

**More detailed comments.**
Lines 30-31.
"Contrary to all other marine-terminating glaciers… decreasing distance to the ice front". I'm not sure that this is always correct, so unless you have a credible reference for the statement, I would suggest that you simplify the sentence to just describe the situation for the two glaciers.
We re-phrased this sentence as: "*In contrast to non-surge-type…*" (line 32).

Re 2.2 'Double-difference interferometry.' Note that any interferometric technique is by definition 'differential'.

Lines 55 – 56. "DInSAR measures phase change between two image acquisitions, which is proportional to displacements in the radar line-of-sight direction." This is strictly not correct; I would prefer something like the following…

Interferometric SAR (InSAR) allows the estimation of the phase change between two image acquisitions which, when the phase is unwrapped (ref), provides data on the difference in slant range to the registered pixels. When the unwrapped phase is then corrected for the different viewing positions and terrain topography (using the baseline and a DEM, add another reference), then the change in the unwrapped phase is proportional to the change in the surface displacement in the radar line-of-sight direction.

I don't think using 'DDInSAR' is necessary. 'DInSAR' is now often used for double differencing interferograms. e.g., Wild et al., 'Differential interferometric synthetic aperture radar for tide modelling in Antarctic ice-shelf grounding zones', The Cryosphere, 13, 3171–3191, https://doi.org/10.5194/tc-13-3171-2019, 2019.

As suggested, we have elaborated on the relation between interferometric phase and surface displacements. This part now reads (starting at line 57): "*DInSAR measures phase change between two subsequent image acquisitions. Once the effects of the satellite viewing geometry and local topography is accounted for (using an external Digital Elevation Model), the unwrapped phase is proportional to displacements in the radar line-of-sight direction (Massonnet et al., 1993).*"

As for the InSAR/DInSAR nomenclature, upon consideration, we agree that introducing the additional abbreviation 'DDInSAR' is not strictly necessary. However, we are also hesitant to use DInSAR (or 'differential SAR interferometry') to refer to the double-difference approach. Although the early/classic literature has arguably not been 100% consistent in this terminology, 'differential interferometry' has previously been used to describe the retrieval of surface displacements (because the process involves differencing of one interferogram with a topographic signal, either from another interferogram or from an external DEM). Hence, we would consider the output of 'DInSAR' to be an interferogram consisting of two repeat-pass SAR retrievals with topography/flat-Earth phase eliminated through baseline/DEM information, meaning that the unwrapped phase is proportional to the motion between the time of the two acquisitions. Double-difference interferometry ('DDInSAR') is then the process of differencing two differential interferograms (i.e. two DInSAR products), which then yields an interferogram whose phase is related to the *difference* in motion between the two time epochs (t1-t2 and t3-t4). Consequently, we have kept this distinction between differential interferograms and double-difference interferograms (although with the omission of the 'DDInSAR' abbreviation).

Line 65. I suggest you drop 'DInSAR'… 'We generate interferograms using …' And line 68; e.g., 'A time series of double difference interferograms is then generated by differencing all temporally neighbouring interferograms.'

Changed as suggested.

2.3 Elevation change measurements.
Line 83. Here you refer to 'accumulation rate' and 'ablation rate'. These are well defined parameters in glaciology neither of which account for height change due to ice motion (in particular along-flow strain rates). So, I suggest you change the wording here, and elsewhere, to make sure that readers understand you are referring to surface height change and height change rate in two areas, one

thickening and the other thinning. In fact, I doubt that the elevation of the upper area is high enough to be in the true 'accumulation zone', i.e., with a positive surface mass balance.

We agree that the use of "accumulation" and "ablation" was confusing, as we are indeed referring to surface elevation changes. Consequently, we have replaced all instances of "accumulation/ablation zone" with either "upper/lower zone" or "thickening/thinning zone" (throughout the text as well as in Figure 1c).

3. Results

You have used two 5 km sections of the NASA ATM flight track over Storstrømmen to monitor height change in what you describe as in the 'accumulation' and ablation' zones. Here you appear to follow the work of Mouginot, et al., who also used this data. Is the 5 km section the same as used in the Mouginot work? Maybe add a comment or reference?

The 5 km sections are centered on the two points used to extract elevation changes in Mouginot et al. (2018). We added the following sentence in line 89, to convey this: "*The 5 km sections are centered on the points used to extract elevation in Mouginot et al. (2018)*". The averaging is intended to lower the influence of spurious noise in the elevation measurements on the resulting elevation change estimates.

Line 100. The noise in the velocity anomaly sub-images for the summer periods is so large that I think you may want to rethink the estimate of summer speed up in... '... ice flow is observed to be quite stable, except for a reoccurring summer speed-up on the order of < 40 m/y (although it should be noted that measurement noise generally increases during summer, likely due to increased surface melt).'

We recognize that the Jun-Aug measurements are indeed very noisy, and concluding a "*summer speed-up on the order of 40 m/y*" is putting too much confidence in these measurements. That being said, we find it likely that the measurements (although admittedly noisy), likely do measure a real speed-up signal (at least locally), based on the fact that a re-occurring summer speed-up has been measured in the past (Vijay et al., 2019).

We have re-formulated the sentence as (lines 105-107): "*An apparent re-occurring speed-up is observed during summer, however, as measurement noise drastically increases during this period (likely due to enhanced surface melt), the confidence in this signal is reduced.*"

Lines 105 – 112. The text here is a mixture of results and conjecture as to the cause of the observed changes. I would prefer that the conjecture re lake drainage, surface speed up based on phase, subsidence, etc., go in the next section. Some more results or clear evidence showing that there are 'lakes' which could periodically release water would be helpful (see below for one example).

As suggested, we have added the additional evidence of lake drainages from ArcticDEM difference maps (please see our response to your final comment). Note that we have also added measurements from another Sentinel-1 track, which justifies our inference of flow speed-up and uplift/subsidence in Figure 3 (see our response under your comment labelled "Figure 3" below). Altogether, we think this provides quite convincing evidence of the inferred lake drainages (justifying presenting them as "observations" more so than conjecture). Although this section still involves some analysis (that could traditionally fit in a Discussion section), we prefer to keep this paragraph in the Results, as it helps us present the various data/evidence in the order that we think is optimal (particularly considering the "Brief Communication" format of the paper).

Lines 120 – 123. '… ice flow accelerations on the order of 50 – 120 m/y,' The units are not appropriate for acceleration, do you mean change in speed of ….
We changed "*acceleration*" to "*speed-ups*" (line 128).

Lines 151 – 154. While I think these statements are correct, the evidence is at best indirect. I think this should be acknowledged.
We changed "*...we find multiple examples of supraglacial and ice-dammed lake drainages…*" to "*...we find evidence of multiple supraglacial and ice-dammed lake drainages…*", to make the claim a bit less assertive (although we would argue that the evidence is quite strong, particularly with the inclusion of ArcticDEM difference maps - cf. our response to your final comment below).

**Figures.**
Figure 1. Good.

Figure 2. The top left sub-image uses a linear colorbar. A non-linear colormap, like Figure 1a would be preferable. A vector showing the zero-Doppler or line-of-sight direction would also be useful here.
We have changed to a non-linear colormap in the top-left panel (the same as in Figure 1a), as suggested. Figure 2 only displays 2D ice velocity magnitude measurements (obtained from the PROMICE products), so a line-of-sight vector is not applicable.

Figure 3. Panels 3c and 3f include labels indicating areas with 'subsidence'. The arguments in the text are somewhat speculative and not proven. I suggest change to 'possible subsidence'.
We acknowledge that the evidence presented in the original version of the manuscript did not justify a confident inference of these vertical signals (as we only show double-difference interferograms from a single track). Therefore, we now provide additional evidence, based on both ascending and descending acquisitions, compiled in the new Figures S2 and S3. We have unwrapped interferograms ("regular" differential interferograms, not double-difference), yielding line-of-sight velocity maps for image pairs spanning the time of suspected flow speed-up and uplift (Fig. S2) and subsidence (Fig. S3). We then subtract a "pre-event" LoS velocity map, to obtain velocity anomaly maps (the right-most panels in Figs. S2-S3). On their own, these LoS velocity anomaly maps can in some cases already indicate the type of displacement: For instance, in Fig. S3 both tracks show LoS anomalies of the same sign (negative) and the vertical sensitivity is roughly equal, while the sensitivity to flow-directed motion is of opposite sign, for the two tracks, indicating that the signals are caused by subsidence. We also apply the inversion/decomposition scheme described in Maier et al. (2023), using the two LoS velocity retrievals to solve for flow-directed and vertical motion anomalies (with the assumption that horizontal flow direction stays constant). This works well in the cases of Fig. S2a-b, Fig. S3a and S3c, as the image pairs are separated by just 0.5 days, but not so well in the case of Fig. S3b, where we only have 12-day image pairs (that are less coherent) separated by 5.5 days. Also, since the flow direction of Storstrømmen approaches perpendicularity with the LoS vectors for parts of the drainage-affected regions, there are gaps in the resulting flow-directed/vertical motion maps. While the decomposed flow-directed/vertical measurements may be more easily interpretable (particularly to readers not familiar with DInSAR), phase unwrapping difficulties and the loss of sensitivity to horizontal motion in certain regions means that we cannot generate a cohesive flow-directed/vertical motion anomaly time series for each of the drainage events. Therefore, we

prefer to show the double-difference interferograms in the main text, as they nicely illustrate the downstream propagation of the dynamic response.

**Supporting supplementary material.**
There are 6 images in the supplementary material. The first three are in support of the text and the interpretation of the results. Figure S4 and S5 were intriguing and led to some investigation using date-specific ArcticDEMs. These supported the interpretation in the communication but provided much more convincing evidence that the glacier speed-up in Fig. S4e was due to significant water outflow from what appears to be a subglacial lake.

This image is from a shaded relief file for the PGC ArcticDEM from 23 March 2019. The red arrow points to the position where the ice surface dropped due to subglacial lake drainage. Note the position of the arrowhead in Fig. S4a pointing to the area of 'inferred drainage' is not accurate and should be changed to point to the correct area. This figure illustrates the DEM from 23 March 2019 after the water outflow from the subglacial lake. This illustrates the height difference between pre- and post-outflow DEMs. The surface falls by up to around 85 m. This complements your observation that the SAR image brightness increased significantly between 11 and 17 Feb. 2019 as the ice broke up due to the surface collapse as the subglacial lake drained. Presumably, the water outflow facilitates the surface ice speed-up seen in S4e and S4f.

Again, we thank the reviewer for taking the time to carry out this extra analysis! Following your advice, we now include ArcticDEM measurements to further support our inference of lake drainage events. We generated DEM-difference maps for all of the events documented in the paper. In all cases (except one), a local drop in elevation could be linked to the investigated events (note that Figure labels below refer to the updated Supplementary Material):

1) 2018 October (Figure 3a-c): Elevation drops from August 16 2018 to April 14 2019 over the ice-dammed lake where Sentinel-1 amplitude changes are also observed;
2) 2022/2023 winter (Figure 3d-f): Elevation drops from May 11 2022 to May 2 2023 over two supraglacial lakes, which also exhibit changes in SAR amplitude;
3) 2019 February (Figure S7): As you demonstrate, an ~80 m elevation drop occurs prior to the observed flow speed-up. However, the drained lake is an ice-dammed lake - this drainage event is actually also documented (based on ICESat-2 measurements) in Figure 2 of this paper (see 'Lake 136030'): https://www.nature.com/articles/s43247-024-01522-4;
4) 2018 December (Figure S6a-c): For this event, we could not confidently identify a corresponding signal in the ArcticDEM difference measurements;
5) 2019 May (Figure S6d-f): Elevation drops from April 4 2019 to July 7 2019 over two supraglacial lakes in upstream Storstrømmen (one of the lakes was also identified as draining in the winter of 2022/2023).

The ArcticDEM difference maps described above have been compiled in a figure (see the new Figure S5), and we point to this added evidence in lines 121-123: "...*and elevation change maps from differencing of individual ArcticDEM strips shows substantial drops in elevation over the same lakes (Figure S5), prompting us to infer their drainage.*"

The arrow in Figure S7a (previously S4a) has been corrected, so that it now points to the correct lake (this was an error in the manual placement of the arrow - good catch!).

As a final note, we discovered a minor error in the date labels of Figures 3, S6, and S7. The acquisition dates of the second image pair in the double-difference interferograms were shifted by 6 days, but have now all been corrected. The error arose from the parsing of dates from the interferogram filenames (in the python script used to generate the plots).

**Summary.**

The brief communication contains interesting material which warrants publication. However, I would like the authors to consider the comments above, and the suggestion to use some time-specific ArcticDEMs. You are welcome to use the above example in your paper, I can provide the files, or you can download e.g., …

SETSM_s2s041_WV01_20190323_1020010082E60600_1020010083934E00_2m_lsf_seg1.tar.gz

From the PGC directory …

https://data.pgc.umn.edu/elev/dem/setsm/ArcticDEM/strips/s2s041/2m/n76w025/

I think the other two files I used are also in this directory (N76W25).

**References**

Massonnet, D., Rossi, M., Carmona, C., Adragna, F., Peltzer, G., Feigl, K., and Rabaute, T.: The displacement field of the Landers earthquake mapped by radar interferometry, Nature, 364, 138–142, https://doi.org/10.1038/364138a0, 1993

Vijay, S., Khan, S. A., Kusk, A., Solgaard, A. M., Moon, T., & Bjørk, A. A. (2019). Resolving seasonal ice velocity of 45 Greenlandic glaciers with very high temporal details. Geophysical Research Letters, 46, 1485–1495. https://doi.org/10.1029/2018GL081503

---

## Author Comment (AC2)

Dear Adrian Luckman,

Thank you very much for your detailed review and constructive comments and suggestions, which we believe have improved the manuscript. Please find below your original comments in black text and our responses in blue (quotes from the updated manuscript are in *blue italics*).

The authors use Sentinel-1 ice velocity products, DinSAR-derived grounding lines, and a variety of surface elevation data sources, to understand build up to the next surge of two important glaciers in Greenland, and document evidence of subglacial drainage events and the dynamic response to them.

This paper brings up to date the work by Mouginot et al., (2018) and provides a small advance in understanding the quiescent phase and predicting the likely year of the next surge. It is well written and very well illustrated and, although it could be considered incremental, I believe it is worthy of publication as a Brief Communication in the Cryosphere subject to some revisions:

GENERAL

1) The paper focusses on SAR and InSAR methods and observations, demonstrating very well developed data analysis skills and figure-making. In contrast, the glaciological discussion (page 6) is rather brief and pays no attention to the literature (surge-related, glacial hydrology-related, or otherwise), which is a potential missed opportunity for influencing the topic and picking up citations. There could be lots to discuss here about where the water goes (ground-water? - this is a growing topic), whether the glaciers are frozen to their beds (see lots of papers about surge initiation, and subglacial water outbursts), and what actually triggers a surge. If the authors do not have the appetite for a literature review, could they co-opt someone (e.g. a well-known surge specialist), to add this extra bit of informed (and referenced) discussion? If not, it may be better to couch this section purely in terms of observations and leave out the under-developed glaciology.

We acknowledge that the discussion of our findings in relation to surge theory was quite brief and lacked context and references. We have revised this section, providing some more context between our findings and the existing literature. The final part of the Discussion now reads (starting at line 158):

*"Using interferometric satellite radar measurements from the past decade, we find evidence of multiple supraglacial and ice-dammed lake drainages, showing that high inputs of water are regularly provided to the subglacial environment. The drainage events all occur outside the melt season, when we would generally expect a less efficient subglacial drainage system and thus a greater increase in basal water pressure, but lead only to transient flow accelerations over timescales of weeks to months. Within the general theory of glacier surges, meltwater inputs to, and subsequent changes in, the subglacial drainage system have frequently been linked to surge initiation (Kamb et al., 1985; Lingle and Fatland, 2003; Dunse et al., 2015; Haga et al., 2020). In a recently proposed generalized surge model based on enthalpy balance, an influx of water to the subglacial system is associated with an increase in enthalpy (Benn et al., 2019, 2022). While the rapid drainage events presented here clearly did not initiate a surge for either glacier, it is possible that similar events may contribute to future surge initiation, once the pre-surge configuration, and thus a state of*

*mass/enthalpy imbalance, has been reached. Alternatively, the external forcing from these episodic, transient inputs of meltwater to the glacier bed may play a lesser role in initiating surges of Storstrømmen and L. Bistrup, which instead may be controlled by a more gradual evolution in basal water pressure and subglacial drainage configuration.*

*A common theory is that surge initiation occurs once enough basal water is accumulated to raise water pressure above ice overburden pressure, enhancing basal motion through sliding and commencing a velocity-frictional heating feedback (Clarke, 1976; Benn et al., 2019). Our observations indicate downstream propagation of water through the subglacial system over timescales of weeks to months, however, it is unclear how much (if any) of this water is stored in the subglacial system. We do note that for several of the identified drainage events, downstream propagation of subglacial water appeared to cease 25 km upstream of the Storstrømmen grounding line (Figs. 3a-c, Fig. S6), suggesting that the drained water volume might not have been fully evacuated. Investigating similar surface-to-bed drainage events (including their frequency) in the time up to and during the next Storstrømmen surge may reveal detailed changes in the subglacial drainage system (in the form of spatial uplift/subsidence patterns - see Figs. S2-S3 - and the temporal propagation of the dynamic response). Continued close monitoring of hydrology-dynamical effects could then help establish the impact of supra- and subglacial drainage events on the surge cycles of Storstrømmen, L. Bistrup Bræ, and other surge-type glaciers."*

While we have made a concerted effort to improve the context and interpretations of our observations, we also recognize that some aspects may have been left out. Ultimately, our goals with this paper are to 1) extend the time series of Mouginot et al. (2018) to update the timing prediction of pre-surge configuration, and 2) to present and share observations of rapid drainage events and highlight the potential of using similar measurements (particularly DInSAR) to study the hydrology-dynamic effects of melt drainage events on surge initiation (and the surge cycle in general).

MINOR

line 50: "Contrary to all other"→ "In contrast to non-surge-type"
Changed to "*In contrast to non-surge-type…*" (line 32).

line 51: "decrease with decreasing distance to the ice front"→ "decrease up-glacier"
Changed to "*...flow speeds decrease up-glacier*" (line 33).

section 2.1: I got very confused over all of the time periods, so I recommend revising this section. Where do the 24-day averaged velocities come from and why are these used to create the quarterly mosaics? Why not go from the natural periodicity (which is unacknowledged as 6-days for part of the time-period) direct to 3-months?
The PROMICE velocity product is distributed as 24-day averaged velocity mosaics (with a 12-day overlap between subsequent mosaics). These 24-day mosaics are generated from individual 6- and 12-day image pairs - we now mention this in line 50 ("*...utilizing all available 6- and 12-day image pairs*"). The additional averaging, down to 3 months, was done for two reasons: 1) to provide

additional noise reduction, and 2) to be able to illustrate a full 2D time series of velocity variation through the entire 2016-2023 period in a somewhat reasonably sized, digestible figure. Ultimately, the goal of including this data/figure is to document that ice velocity in this region is relatively stable, with the exceptions of the summer months (please see our response to your "line 100" comment below) and a few other instances, which we later link to drainage events.

Line 58: "displacement anomalies". This doesn't seem like a suitable term. Anomalies are normally related to long timeseries. I think the term differences or variations is more suitable here.
We rephrased "displacement anomalies" to "displacement changes".

Line 71: Remove "roughly". I'm sure you were as careful as you could be.
Done.

Line 83 (and later): The term "accumulation" here is used to describe vertical uplift, which I guess is the net result of ice inflow, surface snow input and minimal surface melt. So it is not exactly wrong, but might be misinterpreted as simply the surface snow input. I suggest you find another way of expressing this.
We agree that the use of "accumulation" and "ablation" was confusing, as we are indeed referring to surface elevation change. Consequently, we have replaced all instances of "accumulation/ablation zone" with either "upper/lower zone" or "thickening/thinning zone" (throughout the text as well as in Figure 1c).

Line 100: Here I think you are referring to the highly noisy velocity maps for each jun-aug period in figure 2. These appear to be too noisy to make sense, and the apparent (but clearly wrong) signals dwarf those that you are drawing attention to (the non-summer speed-ups). I suggest you either properly filter these data (I am surprised that the PROMISE processing chain has allowed these through), or just remove the jun-aug panels and say that the summer data is not reliable because of surface melt.
We recognize that the Jun-Aug measurements are indeed very noisy, and concluding a "*summer speed-up on the order of 40 m/y*" is putting too much confidence in these measurements. That being said, we find it likely that the measurements (although admittedly noisy), likely do measure a real speed-up signal (at least locally), based on the fact that a re-occurring summer speed-up has been measured in the past (Vijay et al., 2019). As for applying additional filtering or omitting the panels from the Figure, we prefer to show the measurements "as is", as the filtering/culling procedures of the PROMICE product are well described in the Solgaard et al. reference, and then caution the reader that the summer measurements are less reliable (likely due to a heavy influence of surface melt).
We have re-formulated the sentence as (lines 105-107): "*An apparent re-occurring speed-up is observed during summer, however, as measurement noise drastically increases during this period (likely due to enhanced surface melt), the confidence in this signal is reduced.*"

Lines 140-148: Have a rethink of the order of explanation here. It is confusing that you talk about the 2027 date, then the 2040 date, then mention them both again later. This could all be made much clearer with reference to each date, its source, and implication only once.
We have rephrased this paragraph. We want to convey the year in which we expect each of the elevation/grounding line parameters to reach their pre-surge configuration (2027 for the grounding

line and lower zone elevation and 2040 for the upper zone elevation). At the same time, we also wish to point out that, since the grounding line retreat and ice thinning in the lower reservoir seemingly persist at constant rates, the total mass imbalance between the upper and lower zones should continually increase. Therefore, we anticipate that surge initiation is more likely in the earlier parts of the 2027-2040 time frame. The whole paragraph now reads as follows (starting at line 148):

*"Compared to Mouginot et al. (2018), we thus predict that the Storstrømmen grounding line location and lower zone elevation will meet pre-surge (1978) conditions around year 2027 (agreeing well with previous estimates), while mass build-up in the upper reservoir will likely occur later (around year 2040 vs. the previous estimate of 2029-2030), assuming a continuation of current trends (Figure 1c). A presumed requirement for surge initiation is an ice mass imbalance between the upper and lower reservoirs of Storstrømmen (Reeh et al., 1994; Mouginot et al., 2018). Although thickening in the upper reservoir has recently decreased, thinning in the lower zone and retreat of the grounding line appear to persist at steady rates, resulting in a continuous increase in driving stress. Thus, while the precise pre-surge conditions of 1978 are unlikely to be fully reestablished by 2027, surge initiation is anticipated to be more probable in the earlier part of the 2027–2040 time frame. Inferring the timing of a coming surge would provide a valuable opportunity for acquiring in-situ and remote observations in the years up to, during, and after a glacier surge."*

Line 149: "decrease in back pressure". This would be better expressed as "increase in driving stress" Changed to "*continuous increase in driving stress…*" (line 154).

As a final note, we discovered a minor error in the date labels of Figures 3, S6, and S7. The acquisition dates of the second image pair in the double-difference interferograms were shifted by 6 days, but have now all been corrected. The error arose from the parsing of dates from the interferogram filenames (in the script used to generate the plots).

**References**
Benn, D. I., Fowler, A. C., Hewitt, I., and Sevestre, H. (2019). A general theory of glacier surges, Journal of Glaciology, 65, 701–716, https://doi.org/10.1017/jog.2019.62

Benn, D. I., Hewitt, I. J., and Luckman, A. J. (2022): Enthalpy balance theory unifies diverse glacier surge behaviour, Annals of Glaciology, 63, 88–94, https://doi.org/10.1017/aog.2023.23

Clarke, G. K. (1976): Thermal Regulation of Glacier Surging, Journal of Glaciology, 16, 231–250, https://doi.org/10.3189/S0022143000031567

Dunse, T., Schellenberger, T., Hagen, J. O., Kääb, A., Schuler, T. V., and Reijmer, C. H. (2015): Glacier-surge mechanisms promoted by a hydro-thermodynamic feedback to summer melt, The Cryosphere, 9, 197–215, https://doi.org/10.5194/tc-9-197-2015

Haga, O. N., McNabb, R., Nuth, C., Altena, B., Schellenberger, T., and Kääb, A. (2020): From high friction zone to frontal collapse: dynamics of an ongoing tidewater glacier surge, Negribreen, Svalbard, Journal of Glaciology, 66, 742–754, https://doi.org/10.1017/jog.2020.43

Kamb, B., Raymond, C. F., Harrison, W. D., Engelhardt, H., Echelmeyer, K. A., Humphrey, N., Brugman, M. M., and Pfeffer, T. (1985): Glacier Surge Mechanism: 1982-1983 Surge of Variegated Glacier, Alaska, Science, 227, 469–479, https://doi.org/10.1126/science.227.4686.469

Lingle, C. S. and Fatland, D. R. (2003): Does englacial water storage drive temperate glacier surges?, Annals of Glaciology, 36, 14–20, https://doi.org/10.3189/172756403781816464

Vijay, S., Khan, S. A., Kusk, A., Solgaard, A. M., Moon, T., & Bjørk, A. A. (2019). Resolving seasonal ice velocity of 45 Greenlandic glaciers with very high temporal details. Geophysical Research Letters, 46, 1485–1495. https://doi.org/10.1029/2018GL081503

---

## Author Response (AR2)

Dear Stephen Livingstone,

Thank you for serving as editor for this article. As for the technical correction to Line 33, we decided to omit the sentence and slightly re-phrase the preceding/following sentence. The paragraph now reads (starting in Line 31):

"*Storstrømmen and L. Bistrup are currently in a quiescent phase, with both glaciers exhibiting low average flow speeds and limited seasonal fluctuations in comparison to other marine-terminating glaciers in Greenland. For both glaciers, ice is nearly stationary (average flow < 20 m/y) in the last 30-50 km before the ice front, with increasing flow speeds (50 - 200 m/y) further upstream (Figure 1a).*"